# Toward Real Real-Space Refinement of Atomic Models

**DOI:** 10.3390/ijms232012101

**Published:** 2022-10-11

**Authors:** Alexandre G. Urzhumtsev, Vladimir Y. Lunin

**Affiliations:** 1Centre for Integrative Biology, Institut de Génétique et de Biologie Moléculaire et Cellulaire, CNRS–INSERM-UdS, 1 rue Laurent Fries, BP 10142, 67404 Illkirch, France; 2Faculté des Sciences et Technologies, Université de Lorraine, BP 239, 54506 Vandoeuvre-lès-Nancy, France; 3Institute of Mathematical Problems of Biology RAS, Keldysh Institute of Applied Mathematics of Russian Academy of Sciences, 142290 Pushchino, Russia

**Keywords:** real-space refinement, refinement programs, atomic images, map calculation, shell decomposition, inhomogeneous resolution, CPU time

## Abstract

High-quality atomic models providing structural information are the results of their refinement versus diffraction data (reciprocal-space refinement), or versus experimental or experimentally based maps (real-space refinement). A proper real-space refinement can be achieved by comparing such a map with a map calculated from the atomic model. Similar to density distributions, the maps of a limited and even inhomogeneous resolution can also be calculated as sums of terms, known as atomic images, which are three-dimensional peaky functions surrounded by Fourier ripples. These atomic images and, consequently, the maps for the respective models, can be expressed analytically as functions of coordinates, atomic displacement parameters, and the local resolution. This work discusses the practical feasibility of such calculation for the real-space refinement of macromolecular atomic models.

## 1. Introduction

Even though structural biology deals with biological objects of different complexities, sizes, and levels, very impressive results and information have been obtained from macromolecular studies at the atomic level. Two principal methods for such studies, X-ray or neutron crystallography (MX) and cryo-electron microscopy (cryo-EM), describe macromolecular models in terms of positions $r_{n}$, n=1, 2, …, N, of the atomic centers and of the uncertainties of these positions. Below, we discuss only an isotropic uncertainty characterized for each atom by its own atomic displacement parameter (ADP) $B_{n}$. 

The experimental information of these methods is available in different terms. In cryo-EM, the experiment gives the maps of the electrostatic scattering potential $\rho_{obs}\left(r\right)$ as a result of 3D reconstruction from 2D experimentally observed projections. These maps have a limited resolution, which usually varies from one region to another [1]. In MX, the experiment results in a set of Fourier coefficients $F_{obs}\left(s\right)$ of the electron density distribution, or rather the magnitudes $F_{obs}\left(s\right)=\left|F_{obs}\left(s\right)\right|$ of these complex values, which are also known as structure factors. After all, these data are converted into maps $\rho_{obs}\left(r\right)$ of a limited resolution. This procedure consists of several steps, but it is applied only once for a certain period of work and this experimentally based map is used then for validation and the improvement of atomic models. In what follows, we refer to both experimental distributions $\rho_{obs}\left(r\right)$ as a ‘density distribution’. In both cases, this function is considered in a crystal: a real one in MX and a virtual one, containing an isolated macromolecular object per unit cell, in single-particle cryo-EM. 

Atomic models are refined by their best fit to the experimental data; for a recent review, see [2]. For such model-to-data comparisons, model information is expressed in the same terms as the data, $F_{calc}\left(s;\left\{r_{n},B_{n}\right\}\right)$ or $\rho_{calc}\left(r;\left\{r_{n},B_{n}\right\}\right)$, and some score function is calculated. Reducing this function value is expected to be an indicator of a model improvement. Depending on the type of information and respective score functions, structural biologists talk about reciprocal-space refinement
(1)$$\underset{\left\{r_{n},B_{n}\right\}}{\min}f_{reciprocal}\left(F_{calc}\left(s;\left\{r_{n},B_{n}\right\}\right);F_{obs}\left(s\right)\right)$$
and real-space refinement
(2)$$\underset{\left\{r_{n},B_{n}\right\}}{\min}f_{real}\left(\rho_{calc}\left(r;\left\{r_{n},B_{n}\right\}\right);\rho_{obs}\left(r\right)\right).$$

Each of these two types of refinement has its features (e.g., [2,3]) based on the properties of structure factors and density distributions. Each atom, and more generally each piece of the density distribution, contributes to all structure factors. By this reasoning, an appropriate reciprocal-space refinement requires the coordinates and diffraction parameters of all atoms in the unit cell [4], as well as a contribution from all disordered regions, in particular that from the bulk solvent [5]. Differing from this, each atom significantly contributes to the density only in a relatively small region of the space around the atomic center. As a consequence, to calculate an accurate density distribution in the vicinity of an atom, one requires the parameters of only a few atoms: those close to the atom under consideration. Except for the atoms at the molecule surface, especially in the regions partially occupied [6], this density is not influenced by the disordered solvent either [7]. This suggests real-space refinement as the method of choice [8], especially in earlier studies of work when only a partial atomic model is available. Then, the refinement of the inner part of such a model can be performed, ignoring the missed parts of the model and a contribution from the disordered regions, and this step can be completed later with reciprocal-space refinement [3]. 

The minimization of the chosen score function is usually performed iteratively and is ruled by its gradient with respect to the atomic parameters $\left\{r_{n},B_{n}\right\}$. While $\rho_{obs}\left(r\right)$ is obtained only once for a whole refinement procedure in cryo-EM, or is updated from time to time in MX, the maps $\rho_{calc}\left(r;\left\{r_{n},B_{n}\right\}\right)$ are calculated for each model tried during real-space fit.

To obtain model structure factors $F_{calc}\left(s;\left\{r_{n},B_{n}\right\}\right)$ or model density maps $\rho_{calc}\left(r;\left\{r_{n},B_{n}\right\}\right)$ from the atomic parameters, refinement programs use transition modules from one parameter level to the next one. Refinement can require several consecutive transition steps, as Figure 1 shows [9,10]. These calculations should provide sufficiently accurate structure factors or a map and be fast enough to make such calculations useful in practice. An efficient algorithm to calculate the gradient is automatically defined by inverting the scheme of the function calculation [9]. 

The scheme relating different model levels (Figure 1) is hierarchic and direct. This means that one can routinely pass from a previous level of model data to the next one and not necessarily the opposite way. In particular, one can generate a density map on any fine grid from an atomic model and then calculate a set of Fourier coefficients (structure factors) of this grid function. On the contrary, given a set of structure factors, one cannot recover the exact density distribution at any fine grid but only an approximation to it, a map of a limited resolution, calculated as a Fourier series with this set. While the architecture of the reciprocal-space atomic refinement programs is quite established, this is not yet the case for real-space refinement programs. In this work, we discuss the overall scheme and practical steps for such procedures.

## 2. Results

### 2.1. Gaussian Atomic Model

In MX and cryo-EM, the atomic scattering factor is a Fourier transform of a density distribution of an immobile isolated atom placed in the origin, and it is usually approximated by a weighted sum of a few Gaussian functions, $K_{Gauss}\sim 1–5$ [18,19,20,21,22,23]. Coefficients of this sum depend on the diffraction method, given the chemical type of atoms, and eventually on the atomic environment [24]. In what follows, to simplify the illustrations and unless the opposite is written, we consider a ‘movable’ virtual single-Gaussian atom n of a unit ‘charge’ for which its scattering function (structure factor with index s when the atom is placed in the crystal origin, $r_{n}=0$) is equal to
(3)$$F_{n}\left(s;B_{n}\right)=\exp\left[-\frac{\left(b_{A}+B_{n}\right){\left|s\right|}^{2}}{4}\right].$$

Here, $b_{A}$ is the parameter of this immobile Gaussian atom, representing the rate of decrease in the atomic scattering factor with resolution (or, respectively, the width of the peak of the atomic density, as shown below), and $B_{n}$ is its isotropic atomic displacement parameter describing the variation in the position of this atom in time during the experiment or in space over equivalent copies. The density $\rho_{n}^{0}\left(r;B_{n}\right)$ corresponding to this atom is also Gaussian
(4)$$\rho_{n}^{0}\left(r;B_{n}\right)=g\left(r;b_{A};B_{n}\right)={\left(\frac{4\pi }{b_{A}+B_{n}}\right)}^{3/2}\exp\left(-\frac{4\pi^{2}{\left|r\right|}^{2}}{b_{A}+B_{n}}\right).$$

Functions $F_{n}\left(s;B_{n}\right)$ and $\rho_{n}^{0}\left(r;B_{n}\right)$ are spherically symmetric functions decreasing with the distance s=|s| and r=|r| to the origin. The values of $b_{A}$ and $B_{n}$, typical in structural biology, are of order of $10^{1}-10^{2}\;{Å}^{2}$ [25,26]. For $b_{A}+B_{n}\approx 40\;{Å}^{2}$, the value of $\rho_{n}^{0}\left(r;B_{n}\right)$ at |r|=2.5 Å decreases to about 0.002 times the function value in the origin. By this reasoning, when generating a density distribution as a sum of atomic densities, the atomic contributions are cut out beyond $r=\left|r\right|>r_{dens}$ with $r_{dens}\sim 2.5-3.0\;Å$.

### 2.2. Schemes of Reciprocal-Space Refinement

For an atomic model, its structure factors can be directly calculated from atomic coordinates and displacement parameters, isotropic or anisotropic. In this procedure (red arrows in Figure 1), each of N atoms of the model directly contributes to each of the structure factors making, for large macromolecules, the total number of computing operations too high. Modern macromolecular refinement programs obtain these values as Fourier coefficients of the respective density distribution calculated on a regular grid as a function of the model parameters [27,28]. In this two-step procedure (black arrows in Figure 1), the number of operations is independent of the number of structure factors in the first step and is independent of the number of atoms in the second step. The grid size is a common factor which influences the number of operations for both steps. 

The two-step scheme is faster but introduces errors in the calculated values of model structure factors. First, the density is generated within a sphere centered in the atomic position and with the radius $r_{FT}$. For a virtual Gaussian atom (4), the error in the Fourier transform of the atomic density due to this distance cut-off, being expressed with rescaled parameters $x=r/\sqrt{B}$, $X=r_{FT}/\sqrt{B}$, $t=s\sqrt{B}$, and $t=s\sqrt{B}$, is
(5)$$\begin{array}{l} ∆\left(X\right)=\left|F_{exact}\left(t\right)\right|-\left|F_{integral}\left(t;X\right)\right|=\\ \;exp\left(-\frac{t^{2}}{4}\right)-\frac{8\sqrt{\pi }}{t}\int_{0}^{X}\left(2\pi x\right)exp\left(-4\pi^{2}x^{2}\right)sin\left(2\pi xt\right)dx. \end{array}$$

For a given B value, discrepancy (5) is a non-monotonous function of the distance cut-off X (Figure 2) which suggests that the optimally chosen $r_{FT}$ value may be eventually different from $r_{dens}$. The error becomes small for X~0.4, which, for typical B values, means $r_{FT}\sim 2.5-3.5\;Å$. Increasing $r_{FT}$ increases, as a cube, the number $K_{grid}$ of the grid points to which each atom contributes and the CPU time. 

Extra errors in structure factors occur due to the substitution of the integral Fourier transform by the discrete Fourier transform (DFT) using a finite regular grid. Increasing the grid step improves the accuracy but increases $K_{grid}$, again as a cubic function. When using FFT [29], a compromise between accuracy and the computation time has been discussed [28,30]. At conventional resolutions D~2−3 Å, with the standard choice of $r_{FT}\sim 2.5-3.5\;Å$ and the grid step equal to D/4 or D/3, the two steps require CPU time of the same order of magnitude, $T_{density}\sim T_{FT}$. Exact values, e.g., those shown in Section 2.5, also depend on other parameters, for example, the relative unit cell volume per atom [31].

### 2.3. Schemes of Real-Space Refinement

Real-space refinement compares the model map of a density distribution with an experimental one [11]. For an appropriate comparison, the former map should reproduce the imperfections of the latter. The main sources of imperfections of the maps are their limited resolution and an uncertainty in atomic positions. In MX, maps may also be influenced by missed or downweighed reflections. Usually, at the stage of real-space refinement, eventual experimental errors in the map values are neglected. Similar to reciprocal-space refinement, different procedures can be envisaged to obtain a density map from an atomic model.

First, following the principal scheme (Figure 1), given an atomic model, one generates a respective model density and then applies two consecutive Fourier transforms. The grid for the density should be sufficiently fine to assure accurate structure factors. A similar number of calculations is required to obtain a gradient of a real-space score function with respect to the atomic parameters [9]. In total, using this procedure makes real-space refinement more time-consuming than the reciprocal-space one.

Instead, the model map can be calculated directly from an atomic model as a sum of atomic contributions (blue arrows in Figure 1)
(6)$$\rho^{d}\left(r\right)=\overset{N^{atoms}}{\underset{n=1}{\sum }}\rho_{n}^{d}\left(r-r_{n};B_{n},D\right)\;.$$

Here, $\rho_{n}^{d}\left(r-r_{n};B_{n},D\right)$ is no longer an atomic density but its image at a given resolution. To realize such a procedure, one needs to express these images as a function, ideally an analytic one, of the atomic coordinates, isotropic displacement parameter $B_{n}$, and the resolution D. While both increasing the $B_{n}$ value and decreasing the resolution somewhat similarly blur the central peak of the atomic contribution, their effects are different at a distance to the atomic center. Atomic images $\rho_{n}^{d}\left(r;B,D\right)$ are oscillating functions. Their central peak is surrounded by spherically symmetric waves of a decreasing amplitude, known as Fourier ripples.

To avoid the difficulty of modeling atomic images, some programs [32,33,34,35] deal only with the map values in the atomic centers, making the refinement of $B_{n}$ values impossible. Some authors model only the central peak [36,37] or take the exact atomic density instead of its limited-resolution image [11]. To keep the ripples, the atomic images are either precalculated for some grid of $B_{n}$ values [38] or parametrized using a step approximation to scattering functions [39,40].

### 2.4. Map as an Analytic Function

Fourier ripples are the result of the resolution truncation independent of how this truncation has occurred, explicitly or implicitly. The effect of ripples coming from neighboring atoms is prominent at low and medium resolution; moreover, at subatomic resolution, this effect can strongly bias density deformation maps [41]. The amplitude of these ripples decreases, as a function of the distance to the center, much slower than the atomic density itself. The number of atom contributions to a given point increases with the same rate, giving an important cumulative effect of the ripple truncation [42]. For this reason, to calculate the maps accurately, atomic images should include at least a few Fourier ripples before being cut out at some truncation distance $r_{map}$. 

To model oscillating images, Urzhumtsev and Lunin [43] suggested decomposing them into a weighted sum of spherically symmetric terms
(7)$$\Omega \left(x;\mu ,\nu \right)=\frac{1}{\left|x\right|\mu }\sqrt{\frac{1}{4\pi \nu }}\left[\exp\left(-\frac{4\pi^{2}{\left(\left|x\right|-\mu \right)}^{2}}{\nu }\right)-\exp\left(-\frac{4\pi^{2}{\left(\left|x\right|+\mu \right)}^{2}}{\nu }\right)\right]\;.$$

Each such term represents a uniform distribution on the spherical surface of the radius μ blurred with a Gaussian function with a parameter ν. Thanks to the features of function (7), an image of a normalized virtual Gaussian atom (4), placed in the origin, with any value of its atomic displacement parameter $B_{n}$ and at any resolution D is
(8)$$g^{d}\left(r;b_{A};B_{n},D\right)=\frac{4\pi }{3}\overset{M}{\underset{m=1}{\sum }}\kappa^{\left(m\right)}\Omega \left(r;\mu^{\left(m\right)}D,b_{A}+B_{n}+\nu^{\left(m\right)}D^{2}\right).$$

Here, $\mu^{\left(m\right)},\nu^{\left(m\right)},\;\;\mathrm{and}\;\kappa^{\left(m\right)}$ are coefficients of the decomposition of the three-dimensional interference function
(9)$$3\frac{sin\left(2\pi \left|x\right|\right)-\left(2\pi \left|x\right|\right)cos\left(2\pi \left|x\right|\right)}{{\left(2\pi \left|x\right|\right)}^{3}}\approx \overset{M}{\underset{m=1}{\sum }}\kappa^{\left(m\right)}\Omega \left(x;\mu^{\left(m\right)},\nu^{\left(m\right)}\right).$$
into the sum over Ω(x;μ,ν) terms (shell decomposition) [43].

The number M of terms in (9) is defined by $r_{map}$. With (8), the resolution in (6) may be individual for each atomic image, $D=D_{n}$. This value becomes a parameter of an atomic model, characterizing how confidently $B_{n}$ and $r_{n}$ values are found from the given map. When an atomic density $\rho_{n}^{d}\left(r,B,D\right)$ is represented by a few Gaussians, its image is a respective weighted sum of (8), one per Gaussian.

We illustrated this latter option with Figure 3, which shows a simulated inhomogeneous-resolution map. This map is directly calculated as (6)–(8) for a protein model of IF2 [44] placed in a virtual unit cell in space group P1, similar to cryo-EM models. Here, the resolution was artificially assigned as 2 Å in the center of the molecule, increasing, as a function of distance, up to 5 Å at its periphery. 

Actually, the shell decomposition into a sum of term (7) can be applied to any spherically symmetric oscillating function in space. In particular, an atomic image at a given resolution D for any $B_{n}$ can be directly represented as
(10)$$\rho_{n}^{d}\left(r;B_{n},D\right)=\frac{4\pi }{3}\overset{M}{\underset{m=1}{\sum }}C^{\left(m\right)}\Omega \left(r;R^{\left(m\right)},B_{n}+B^{\left(m\right)}\right).$$
where the coefficients $R^{\left(m\right)}$, $B^{\left(m\right)}$, and $C^{\left(m\right)}$ are calculated for an immobile atom and are universal for all atoms of the given chemical type. Representation (10) reduces the number of terms in comparison with (8) and (9), and thus accelerates calculations while the resolution becomes no more variable.

### 2.5. Comparison of Schemes of Real-Space Refinement

Similar to the structure factor calculation, now we have two ways to obtain an accurate model map: a step-by-step numeric and a direct analytic. The former consists of three steps. First, $T_{density}$ time is required to calculate the exact density distribution on a regular and sufficiently fine grid, with each atom contributing within a sphere of a given radius $r_{FT}$. Second, FFT is applied to this function requiring $T_{FFT\_SF}$ time. Finally, one more FFT is applied to the obtained Fourier coefficients to produce a map of the required resolution on a regular grid, which is usually coarser than the initial one, requiring $T_{FFT\_map}\leq T_{FFT\_SF}$. The map errors become unacceptably large when a too large step $h_{density}$ of the initial grid or a too short $r_{FT}$ are taken. For conventional resolutions  D~2−3 Å, the standard values are $h_{density}\sim D/3-D/4\;Å$ and $r_{FT}\sim 2.5-3.5\;Å$ with no need for artificial manipulations with displacement parameters and increased $r_{FT}$, which may be required for lower resolutions [28,30]. 

The final map is calculated with the same step as the experimental one, usually $h_{map}\sim D/2-D/3\;Å.$ The alternative, direct map calculation consists of a single step requiring CPU time $T_{direct}$. This value depends on the grid step $h_{map}$ of the map, the same as above, and on the truncation distance $r_{map}$ for the atomic images. To obtain accurate maps, this distance has been recommended to be kD/2, with k equal to 4 or 5, or a higher integer [42]. The sum over Ω(x;μ,ν) should include the terms significantly contributing up to this distance. 

To compare the computational efficiency of the two ways to calculate the model maps, we made a numeric experiment with the IF2 model [44] placed in a virtual unit cell with the sides 80 × 120 × 100 Å in space group P1 remining a cryo-EM case. A conventional five-Gaussian approximation to the atomic density was used [23]. We made calculations at the resolution of 2 Å, with varying grid steps and truncation radii. We used the original crystallographic FFT program [46] and our own fast-written rather non-optimized programs to obtain the model density distributions and to calculate directly the limited-resolution maps. The CPU time varies with the computer, compiler, and degree of the algorithm optimization. Additionally, for the same grid, $T_{density}$ and $T_{direct}$ can vary if the model contains more or fewer atoms. This means that when comparing $T_{density}+T_{FFT\_SF}+T_{FFT\_map}$ with $T_{direct}$, as obtained below, some margins should be considered. 

Figure 4 shows CPU time, as a function of the grid step and truncation radius, for the components of the three-step map calculation. $T_{density}$ is near proportional to the number of Gaussians in the approximation. For the given example, using a single-Gaussian approximation, not used in practice, reduces the respective values by four (not shown).

Figure 5 shows CPU time, as a function of the grid step and truncation radius $r_{map}$, for the direct map calculation. We calculated the map for both options, when the resolution is fixed and the simplified decomposition (10) is used, and for the variable-resolution option. The latter multiplies CPU time roughly by four, as it was for the density calculation with multi-Gaussians. One should note that when increasing $r_{map}$ from 4 Å to 5 Å, we not only increase the number of grid points to which each atom contributes but also increase the number of terms in (8) and (10). Inversely, we reduce one term when shortening $r_{map}$ to 3 Å, the distance of which is not recommended except at early refinement iterations [42].

## 3. Discussion

Figure 6 compares CPU time for the different sets of parameter values eventually applicable in practice, i.e., giving sufficiently accurate maps while not requiring excessive time. The results are shown when the resulted map is calculated on the grid with the step D/2, D/3, or D/4; the last group is not expected to be used at the refinement step and is given for reference.

The direct map calculation gives the results roughly for the same time or faster than the multi-step procedure, even when this gain is not of an order of magnitude. Using the fixed-resolution image decomposition, especially with $r_{map}=4\;Å$ for the chosen resolution of 2 Å, is advantageous and can be used as a default option for real-space refinement. Using $r_{map}=5\;Å$ at the final refinement iteration is also acceptable and recommended. 

A particularly important property of the suggested procedure is the possibility to routinely calculate the maps of an inhomogeneous resolution from atomic models (Figure 3). Figure 6 shows that such calculation still be computationally efficient when using $r_{map}=4\;Å$ and the output grid step D/2. Theoretically speaking, maps calculated on such grids contain all information which is contained in the maps with a finer grid and, therefore, may be sufficient for real-space refinement. Calculations of an inhomogeneous-resolution map on a finer grid or with a larger truncation radius may make $T_{density}$ larger than the total time of the three-step calculation, but this is the price for the possibility to introduce and refine individual atomic resolution $D_{n}$, and Figure 6 shows that this price is not excessive. 

From a qualitative point of view, the mathematical features of (7) lead to a new concept when the local resolution is associated with atoms. As a consequence, it can be included into the list of the parameters to be refined and reported as the result of real-space refinement. A feature of the particular map used for refinement is that it characterizes the confidence of the atomic parameters. Another important point is that the discrepancy between the experimental and the model maps becomes an analytic function of all these parameters, and all necessary partial derivatives required for real-space refinement become analytic functions as well. 

Concluding, the features discussed above make the real-space refinement of atomic coordinates and atomic displacement parameters feasible without appealing to reciprocal-space data and tools.

## Figures and Tables

**Figure 1 ijms-23-12101-f001:**
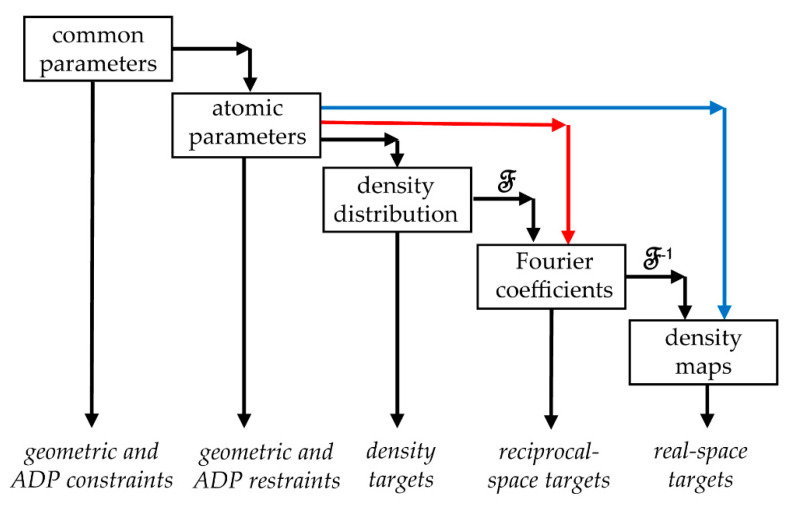
Levels of macromolecular parameterization in MX and cryo-EM. By ‘density distribution’, we consider various kinds of scalar functions in space, such as an electron or nuclear scattering density distribution in crystallography or scattering electrostatic potential in cryo-EM, etc. The term ‘density map’ stands for maps of any of these distributions. Atomic parameters are usually the coordinates of the centers of atoms and their displacement parameters, ADP. Common parameters may be dihedral angles [11,12,13], rigid-body parameters [14], common ADP values for all atoms of the residue [15] or TLS parameters [16,17], or something else, describing common features of an atomic group. Black arrows show the step-by-step hierarchic recalculation of the model parameters; the red and blue arrows illustrate alternative direct calculations of structure factors and maps from model parameters.

**Figure 2 ijms-23-12101-f002:**
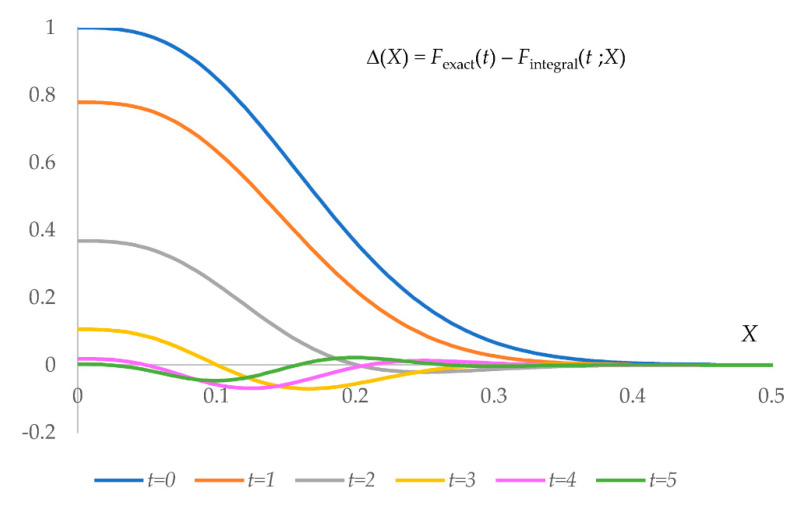
Error in the Fourier transform of a density of a Gaussian virtual atom. Error ∆(X) is given as function (5) of the dimensionless truncation radius, $X=r_{FT}/\sqrt{B}$, and for different values of the parameter $t=s\sqrt{B}$. ∆(0) is equal to the exact value |F(s;B)| for the respective $s\sqrt{B}$.

**Figure 3 ijms-23-12101-f003:**
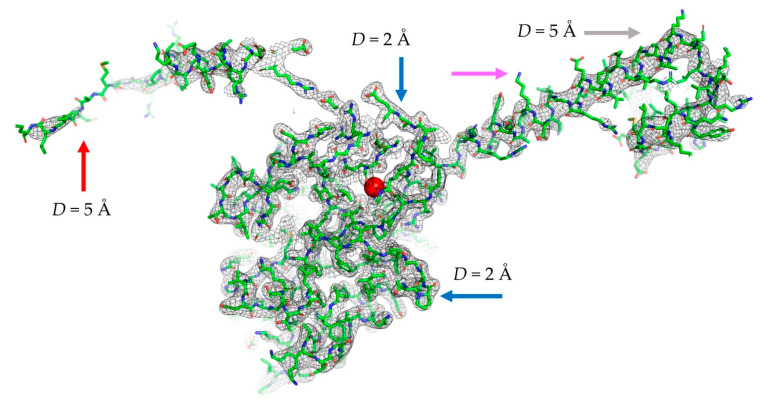
Map of an inhomogeneous resolution calculated in a single run. Map resolution varies from 2 Å around the molecular center (red sphere) to 5 Å at the periphery. Color arrows indicate the regions of a high resolution and small ADP (blue), high resolution and large ADP (magenta), low resolution and small ADP (grey), and low resolution and large ADP (red). Figure has been prepared using Pymol [45].

**Figure 4 ijms-23-12101-f004:**
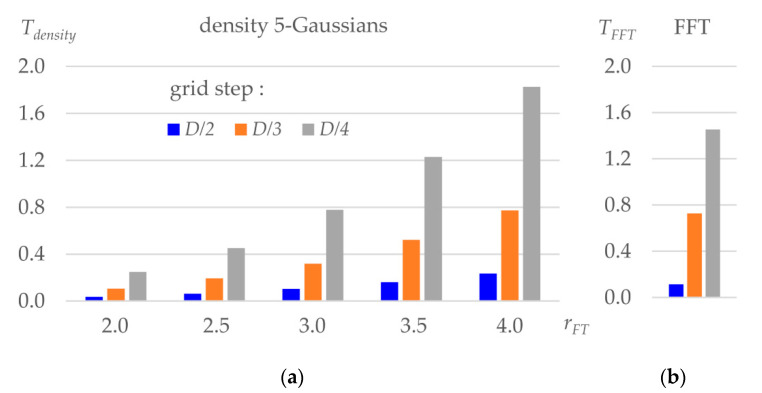
CPU time for the three-step map calculation for different grid steps expressed as a part of the resolution *D*. (**a**) CPU time, in seconds, to calculate a density distribution for the test protein model using different truncation distance $r_{FT}$. (**b**) CPU time to calculate FFT on a grid as defined in (a).

**Figure 5 ijms-23-12101-f005:**
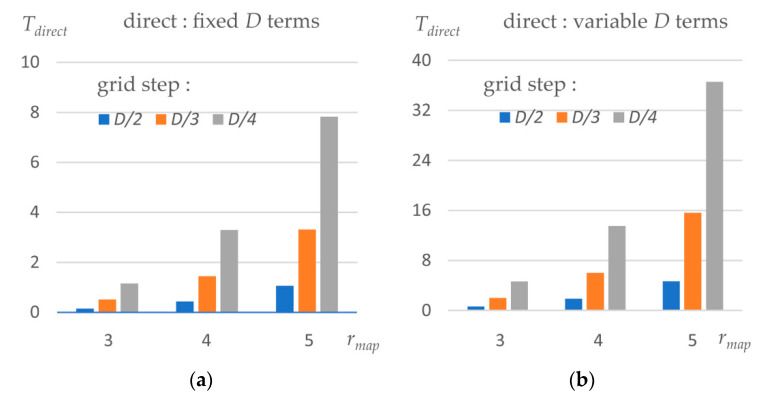
CPU time for the single-step map calculation, as a function of the truncation distance and the grid step, expressed as a part of the resolution *D*. (**a**) CPU time, in seconds, to calculate a density map for the test protein model using the simplified decomposition (10) of atomic images at a fixed resolution. (**b**) The same using the variable-resolution terms (8).

**Figure 6 ijms-23-12101-f006:**
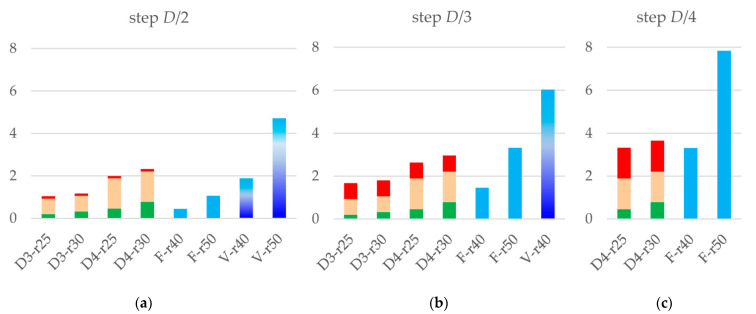
CPU time, in seconds, for different values of parameters used to calculate the model map. Multicolor columns represent the multi-step calculation with the green part for $T_{density}$, beige for $T_{FFT\_SF}$, and red for $T_{FFT\_map}$. Index ‘D’ indicates the grid step for the density, as a part of the resolution, *D/3* or *D/4*; index ‘r’ value is equal to the truncation distance times ten. Blue and variable-blue columns stand for $T_{direct}$ for the fixed-resolution and variable-resolution decompositions, respectively, as indicated by ‘F’ and ‘V’ letters. The grid step of the resulted model map is equal to (**a**) *D/2*; (**b**) *D/3*; and (**c**) *D/4*.

## Data Availability

The data and programs used for tests are available by request from the authors.

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
