# Peer review of "Toward Real Real-Space Refinement of Atomic Models"

_ijms, 2022, doi:10.3390/ijms232012101_

Round 1
Reviewer 1 Report
I find this paper to be well presented overall. The authors' goal to
contribute "Toward real real-space refinement of atomic models" is achieved, making the contribution of this work appealing for a broad audience given its practicality.
I do suggest the authors to consider revising the paragraph starting at line 264 as I found it a bit confusing. For example,
"From the qualitative point of view, mathematical features of (7) lead to a new concept when the local resolution is associated with atoms, is included into the list of the parameters to be refined and reported as the result of real-space refinement."
The verb "is included" (after "atoms") refers to what?
My understanding goes more on what follows (but I might be wrong, so please clarify)
"From the qualitative point of view, mathematical features of (7) lead to a new concept when the local resolution associated with atoms is included into the list of the parameters to be refined and reported as the result of real-space refinement."
Also on the next part of the paragraph
"Discrepancy between the experimental and the model maps becomes and analytic function of all these parameters, and all necessary partial derivatives required for real-space refinement become analytic functions as well"
I think the word after "becomes" should be "an" rather than "and".
Author Response
Open Review 1
I do suggest the authors to consider revising the paragraph starting at line 264 as I found it a bit confusing. For example,
"From the qualitative point of view, mathematical features of (7) lead to a new concept when the local resolution is associated with atoms, is included into the list of the parameters to be refined and reported as the result of real-space refinement."
The verb "is included" (after "atoms") refers to what?
My understanding goes more on what follows (but I might be wrong, so please clarify)
"From the qualitative point of view, mathematical features of (7) lead to a new concept when the local resolution associated with atoms is included into the list of the parameters to be refined and reported as the result of real-space refinement."
Corrected in the revised version
Also on the next part of the paragraph
"Discrepancy between the experimental and the model maps becomes and analytic function of all these parameters, and all necessary partial derivatives required for real-space refinement become analytic functions as well"
I think the word after "becomes" should be "an" rather than "and".
Corrected in the revised version

Reviewer 2 Report
The real-space refinement of macromolecules is a well established technique in the field. Yet there is no mention of widely used programs, i.e. COOT or Phenix for performing these tasks. How does your proposed method compare with the methods used in these and other similar programs?
(16) They can be expressed analytically as functions of coordinates, atomic displacement parameter ...
The use of “They” is not definitive. Does it refer to the “atomic images”? In which case the sentence would be clearer if it began “These atomic images ...”.
(48) Last-years development of experimental techniques positions real-space refinement as the method of choice for modern studies .
This statement is a little bit ambiguous. Real-space refinement is key to structure building and refinement in Cryo-EM maps, and an important part of model-building in MX.
Figure 1
What is the significance of the line colouring? Defined on lines 99, 104, 145? Figure and caption should be self explanatory.
Equation 3
What does the parameter bA represent?
(93) For bA + Bn ≈ 40 Å2, the value of ρ(?;Bn) at |?| = 2.5 Å drops by about 0.002 times in ...
The Gaussian atomic density either: “decreases by a factor of 500” or “decreases to about 0.002 times its initial value”.
Figures 4,5,6
These figures all relate to the computational time required to perform operations. They are better suited to the Supporting Information section and summarized in the body of the paper.
Author Response
Open Review 2
The real-space refinement of macromolecules is a well established technique in the field. Yet there is no mention of widely used programs, i.e. COOT or Phenix for performing these tasks. How does your proposed method compare with the methods used in these and other similar programs?
We inserted the references on the principal approaches to real-space refinement, with and without modeling atomic density.
(16) They can be expressed analytically as functions of coordinates, atomic displacement parameter ...
The use of “They” is not definitive. Does it refer to the “atomic images”? In which case the sentence would be clearer if it began “These atomic images ...”.
Corrected in the revised version.
(48) Last-years development of experimental techniques positions real-space refinement as the method of choice for modern studies.
This statement is a little bit ambiguous. Real-space refinement is key to structure building and refinement in Cryo-EM maps, and an important part of model-building in MX.
Certainly, the reviewer has a point but we cannot fully agree with this. Except of use for model building in programs such Coot, the real-space refinement, the first macromolecular refinement procedure, was under-used in MX exactly because of missed respective tools. In particular, in MX it can be very important for crystals with large unmodeled parts which will bias reciprocal-space refinement of the partial model. We include a reference for an external opinion.
Figure 1
What is the significance of the line colouring? Defined on lines 99, 104, 145? Figure and caption should be self explanatory.
Corrected in the revised version.
Equation 3
What does the parameter bA represent?
Explained in the revised version.
(93) For bA + Bn ≈ 40 Å2, the value of ρ(?;Bn) at |?| = 2.5 Å drops by about 0.002 times in ...
The Gaussian atomic density either: “decreases by a factor of 500” or “decreases to about 0.002 times its initial value”.
Corrected in the revised version.
Figures 4,5,6
These figures all relate to the computational time required to perform operations. They are better suited to the Supporting Information section and summarized in the body of the paper.
These figures visualize the answer on the principal issues discussed in the manuscript: while the suggested approach is theoretically justified, whether it can be used in practice due to changes of parameters in comparison with those used for the density calculations. Summarize these results in the body of paper and explain them by full text does not seem to be optimal from our point of view.

Reviewer 3 Report
The manuscript by Urzhumtsev and Lunin describes a fundamentally new method to calculate X-ray and cryo-EM maps. I have the impression that this manuscript is meant as the “official” publication of the following preprints:
doi: https://doi.org/10.1101/2022.03.28.486044
https://ui.adsabs.harvard.edu/link_gateway/2022arXiv220608935U/arxiv:2206.08935
The cryo-EM technology has recently undergone a quantum leap in performance, leading to the determination of a large number of macromolecular structures that could not be solved by crystallography. The computational methods to process cryo-EM data and refine structures still have to catch-up. Improvements are very needed and many research groups are actively working to improve the cryo-EM software. The current manuscript is part of these efforts.
As mentioned above, the program describes a new method to calculate model density maps, by describing atoms as symmetric oscillation functions in space. As far as I can see, a connection to a refinement program has not yet been made.
While I believe that this new method could be an important contribution to the field, there are some points that are not clear to me:
1) The role of this program in MX. In MX, the primary experimental result is a diffraction data set in reciprocal space, making refinement in reciprocal space the first choice. Besides a model map, produced with the new method, an “experimental” map is needed and here I see no way around the 3-4 step conventional procedure to calculate this one. If the authors think their method could be useful for MX as well, they should explain more clearly why and how.
2) How is the map organized? From the manuscript, I get the impression that it is a (long) list of functions, remotely reminiscent of the Fourier terms in diffraction data, but it could also be a grid. The authors should explain how their maps look like.
3) Fourier ripples. This puzzles me a little bit. Fourier ripples arise when a Fourier series is truncated at a point where the Fourier terms still have a significant value. When the series is truncated at a point where the Fourier terms are negligible, no ripples should arise. Nowadays, MX data sets are cut at CC1/2 is 0.5 or 0.3 and Fourier ripples are no longer an issue. For cry-EM, the primary experimental result is a map, which should include all available Fourier terms, unless the aperture is a problem. It could also be that map sharpening is causing truncation effects. Here the authors should explain where the ripples come from.
4) Local resolution is correlated to B-factors. Regions with high B-factors will have a low local resolution. Except for Fourier truncation effects, all local resolution effects can be completely encoded in the B-factors. However, refinement of B-factors at (very) low resolution is tricky, and here the local resolution may have advantages. Also the density calculated by the new method may have a more similar scale for regions with high and low local resolution, making analysis and model building easier. If the authors think differently, they should more clearly explain why local resolution cannot be encoded in B-factors. It would be good to add to figure 3 a picture of the same protein, with a map calculated using conventional methods. My bet would be that they would look very similar, but I may be mistaken.
Some other points:
Page 1, line 40-41 “for a recent review, see, for example, [3]”. Reference 3 is from 1985. I would not call a 37 years old review recent and would delete “recent” from this phrase.
Page 6; Figure 3: Which protein (PDB code) was used to generate this picture? For comparison, the authors should add a picture of the same protein with a map, calculated using conventional methods.
In summary, the model map calculation program is interesting, but without a connection to a refinement program merely of theoretical interest.
Author Response
Open Review 3
The manuscript by Urzhumtsev and Lunin describes a fundamentally new method to calculate X-ray and cryo-EM maps. I have the impression that this manuscript is meant as the “official” publication of the following preprints:
doi: https://doi.org/10.1101/2022.03.28.486044
https://ui.adsabs.harvard.edu/link_gateway/2022arXiv220608935U/arxiv:2206.08935
This is true but partially. This project contains a large number of facets, from pure mathematics to applications, and this manuscript addresses one of them, rather complementary to the preprints cited.
The cryo-EM technology has recently undergone a quantum leap in performance, leading to the determination of a large number of macromolecular structures that could not be solved by crystallography. The computational methods to process cryo-EM data and refine structures still have to catch-up. Improvements are very needed and many research groups are actively working to improve the cryo-EM software. The current manuscript is part of these efforts.
As mentioned above, the program describes a new method to calculate model density maps, by describing atoms as symmetric oscillation functions in space. As far as I can see, a connection to a refinement program has not yet been made.
True
While I believe that this new method could be an important contribution to the field, there are some points that are not clear to me:
- The role of this program in MX. In MX, the primary experimental result is a diffraction data set in reciprocal space, making refinement in reciprocal space the first choice. Besides a model map, produced with the new method, an “experimental” map is needed and here I see no way around the 3-4 step conventional procedure to calculate this one. If the authors think their method could be useful for MX as well, they should explain more clearly why and how.
We introduced some clarifications as required. We remind a difference in the way how the maps are obtained in EM and MX. It is true that the MX “experimental” maps, except rare cares, are not fully based on the experimental data. Our approach does not address their calculation of such “experimental” map calculated only once for a cycle of model refinement. On the contrary, real-space refinement requires multiple calculations of a map form an atomic model and we suggest how to calculate such maps.
- How is the map organized? From the manuscript, I get the impression that it is a (long) list of functions, remotely reminiscent of the Fourier terms in diffraction data, but it could also be a grid. The authors should explain how their maps look like.
A map, calculated from a model, is presented, as usually, as a 3D array of real numbers. And the procedure we discuss in the manuscript suggests how to calculate these numbers.
- Fourier ripples. This puzzles me a little bit. Fourier ripples arise when a Fourier series is truncated at a point where the Fourier terms still have a significant value. When the series is truncated at a point where the Fourier terms are negligible, no ripples should arise. Nowadays, MX data sets are cut at CC1/2 is 0.5 or 0.3 and Fourier ripples are no longer an issue. For cry-EM, the primary experimental result is a map, which should include all available Fourier terms, unless the aperture is a problem. It could also be that map sharpening is causing truncation effects. Here the authors should explain where the ripples come from.
The Fourier ripples appear when a function is substituted by a Fourier series truncated as some limit, e.g. by missed diffraction data or by FSC-correlation. Talking about maps of a limited resolution means that there is such resolution cut-off applied explicitly or explicitly. We add a comment that while the amplitude of these ripples, coming from a limited-resolution image of individual atom, decrease with a cut-off distance, the number of atoms contribution to a given point, increases respectively. In overall, this gives important cumulative effects.
4) Local resolution is correlated to B-factors. Regions with high B-factors will have a low local resolution. Except for Fourier truncation effects, all local resolution effects can be completely encoded in the B-factors. However, refinement of B-factors at (very) low resolution is tricky, and here the local resolution may have advantages. Also the density calculated by the new method may have a more similar scale for regions with high and low local resolution, making analysis and model building easier. If the authors think differently, they should more clearly explain why local resolution cannot be encoded in B-factors. It would be good to add to figure 3 a picture of the same protein, with a map calculated using conventional methods. My bet would be that they would look very similar, but I may be mistaken.
We included respective comments in the text. The effects of these two phenomena are similar in blurring the central peak but are different far of it. As explained above, Fourier ripples for the resolution truncation have a significative cumulative effect that is absent for blurring the images with a Gaussian function.
Some other points:
Page 1, line 40-41 “for a recent review, see, for example, [3]”. Reference 3 is from 1985. I would not call a 37 years old review recent and would delete “recent” from this phrase.
Our excuses; this was a result of a last-minute correction in the list of references. Corrected.
Page 6; Figure 3: Which protein (PDB code) was used to generate this picture? For comparison, the authors should add a picture of the same protein with a map, calculated using conventional methods.
We corrected the message in the revised version. Here, we only illustrate how computationally a map of a variable (inhomogeneous) resolution can be generated and a result of such a procedure.
In summary, the model map calculation program is interesting, but without a connection to a refinement program merely of theoretical interest.
We agree that this and relevant works prepare the respective theory and algorithms to be integrated into refinement programs and eventually in some other projects which can be done by program developers independently or (for example, the Phenix team) with our collaboration.

Round 2
Reviewer 2 Report
Much improved presentation. The language still needs improving in some sections for clarity.
Figure 5. The images in a and b seem to be duplicated top/bottom.
(319) This is not a complete sentence or thought.
As a consequence, such feature makes it possible ‘real’ real-space refinement of atomic coordinates and atomic displacement parameters, with a reference to reciprocal-space refinement tools neither for model-to-data comparison nor in intermediate steps of the map calculation.
Author Response
Open Review 2
The language still needs improving in some sections for clarity.
Answer : We re-read the text and tried to improve found errors.
Figure 5. The images in a and b seem to be duplicated top/bottom.
Answer : The Figure was correct; it seems to be an effect of unmasking corrections
(319) This is not a complete sentence or thought.
As a consequence, such feature makes it possible ‘real’ real-space refinement of atomic coordinates and atomic displacement parameters, with a reference to reciprocal-space refinement tools neither for model-to-data comparison nor in intermediate steps of the map calculation.
Answer : We agree with the remark and rewrote this phrase.

Reviewer 3 Report
The authors have addressed most of my comments, although I am still not convinced that there will be significant Fourier ripples when a Fourier series is cut at a point where the amplitudes have a negligible value.
However, in their update, the authors introduced a bold statement where I strongly disagree:
(Page 2, lines 58-60) “In particular, in MX real-space refinement allows one to routinely proceed with incomplete models, including those with the unmodeled bulk solvent component or missed domains.”
If the “observed” density map is calculated on the basis of an incomplete model, which is usually the case, the resulting map will have extremely poor density for the missing parts, making proceeding routinely impossible whether using conventional or reciprocal space refinement. It may be that reference [43] explains this but references 24-43+ are missing in the revised manuscript I received, so I cannot check this. However, my guess is, that this reference refers to cryo-EM.
Unless the authors can provide convincing evidence for this, they should remove this statement.
The revised manuscript appears to be hastily put together, especially with half of the references missing. The authors should go carefully through the manuscript to make sure that no artefacts of previous edits are left.
Author Response
Open Review 3
|
|
|
|
|
|
The authors have addressed most of my comments, although I am still not convinced that there will be significant Fourier ripples when a Fourier series is cut at a point where the amplitudes have a negligible value.
Answer : The problem of Fourier ripples is sometimes underestimated. The effect of ripples coming from neighboring atoms is prominent at can be seen at a low resolution. Numeric tests performed at a medium resolution demonstrated that the cumulative effect of ripples coming to the center of an atom from its neighbors distanced to 10 Å may reach 10%. This is small for a visual analysis, and not for an accurate refinement. At subatomic resolution, such as 0.6-0.9 Å, the effect of ripples can be observed in difference Fourier maps and confused (which had happened in practice) with the deformation density. As example can be found at Fig. 4, in Afonine et al., 2004 https://doi.org/10.1107/S0907444903026209. We have added some comments and the reference to the revised version.
However, in their update, the authors introduced a bold statement where I strongly disagree:
(Page 2, lines 58-60) “In particular, in MX real-space refinement allows one to routinely proceed with incomplete models, including those with the unmodeled bulk solvent component or missed domains.”
Answer : We have rewritten this part to make the statement more accurate and explained to which degree it is valid.
The revised manuscript appears to be hastily put together, especially with half of the references missing. The authors should go carefully through the manuscript to make sure that no artefacts of previous edits are left.
Answer : Unfortunately, this happened due a numerous modifications introduced in the revised version and which were improperly masked by Word; we are sorry for this. Our own text, as we checked it before sending, contained all references shown properly. This time again, we checked this on our side for this second revision.
